# BNT162b2 vaccine uptake and effectiveness in UK healthcare workers – a single centre cohort study

Tariq Azamgarhi [1✉], Michelle Hodgkinson[2], Ashik Shah[1], John A. Skinner[3,4], Iva Hauptmannova[3], Tim W. R. Briggs[5] & Simon Warren [6,7✉]

In this single centre cohort study we assessed BNT162B2 vaccine uptake and effectiveness among UK healthcare workers (HCWs) during a time of high community COVID-19 prevalence. Early uptake among HCWs was 62.3% (1409/2260), however there were significant differences in uptake between age groups, ethnic origins, and job roles. Uptake increased to 72.9% after a vaccine hesitancy working group implemented specific measures. In the 42 days after vaccination, 49 new cases of COVID-19 were identified, of which 7 (14.3%) occurred in HCWs who were beyond 10 days of vaccination. Kaplan–Meier curves for partially vaccinated and unvaccinated groups were congruent until day 14 and continued to diverge up to 42 days. Cox regression analysis showed a 70.0% (95%CI 6.0–91.0; p=0.04) risk reduction for COVID-19 infection in partially vaccinated HCWs. Here we report early vaccination rates among HCWs are generally high although uptake is lower in certain groups. It is possible to improve vaccine uptake and efforts should focus on this, however, significant resource is required. The BNT162B2 vaccine is effective from 14 days post-vaccination in a frontline clinical setting and protection continues beyond 21 days post 1st dose without a 2nd dose, being given.

[1] Pharmacy Department, The Royal National Orthopaedic Hospital NHS Trust, Stanmore, Middlesex, UK. [2] Human Resources, The Royal National Orthopaedic Hospital NHS Trust, Stanmore, Middlesex, UK. [3] Research and Innovation Department, The Royal National Orthopaedic Hospital NHS Trust, Stanmore, Middlesex, UK. [4] University College London, London, UK. [5] Get it Right First Time, The Royal National Orthopaedic Hospital NHS Trust, Stanmore, Middlesex, UK. [6] Bone Infection Unit, The Royal National Orthopaedic Hospital, Stanmore, Middlesex, UK. [7] The Royal Free Hospital NHS Foundation Trust, Hampstead, London, UK. ✉email: tariq.azamgarhi@nhs.net; simon.warren@nhs.net

COVID-19 has had a significant impact on populations worldwide. Healthcare workers (HCWs) have been disproportionately affected and have therefore been designated a high priority group for vaccination by the UK Joint Committee on Vaccination and Immunisation[1]. Vaccines have been shown to be highly effective in clinical trials, with several now approved for clinical use[2–4] although uptake and effectiveness in an NHS healthcare setting has not yet been described. The dosing schedule for the BNT162B2 vaccine is a single 30-mcg dose at day 0 and repeated at a minimum of 21 days but due to limited availability, UK government strategy has been to delay the second dose until 3 months[5]. In this single-centre cohort study in London we evaluate uptake and effectiveness after a single dose of the BNT162b2 vaccine during a time of high community COVID-19 prevalence.

## Statistical analysis

We summarised data using descriptive statistics. Categorical data on baseline characteristics were compared using a two-sided Pearson's $\chi^2$ test.

We applied survival analysis to investigate the impact of COVID-19 vaccination on new COVID-19 cases. The outcome variable was time to infection, constructed as the time between vaccination and symptom onset or time of first positive test if asymptomatic. A start date of 15 January 2021, coinciding with the end of the RNOH vaccination programme, was used for partially vaccinated HCWs. All HCWs not infected on 26 February 2021 were censored. HCWs who had received the BNT162B2 vaccine elsewhere were included in the analysis. HCWs who reported to have received a different COVID-19 vaccine at follow-up were excluded. The Kaplan–Meier method was used to plot cumulative hazards for partially vaccinated and unvaccinated groups. In the phase 2/3 safety and efficacy study of the BNT162B2 vaccine, it was demonstrated that COVID-19 vaccination would unlikely have an impact on COVID-19 infections until day 14 post vaccination[3]. Statistically, this means that the hazards are unlikely to be proportional. There may be expected to be no effect in the first 13 days, followed by an effect after this. To allow for the likely non-proportional hazards, two

sets of analyses were performed. The first set compared the groups from day 0 to day 13. A second set of analyses compared the groups with the start point being day 14, up until the end of the follow-up period. As it is unlikely that HCWs would have had more than one COVID-19 infection in such a short space of time, HCWs with a COVID-19 infection within the first 13 days, in both groups, were omitted from the analysis of the second period. Due to the survival nature of the outcome, the analyses were performed using Cox regression for both time periods. Initially a simple 'unadjusted' comparison between partially vaccinated and unvaccinated groups was made. Subsequently, the groups were compared adjusting for demographic details found to vary significantly between groups. Hazard ratios were also adjusted for underlying COVID-19 infection rates in the London area (Fig. 1). This was treated as a time-varying covariate, with different values for each day of the follow-up period. Statistical analysis was performed using SPSS version 25.0 (IBM, Chicago, IL).

## Results

In total, 1409 (62.3%) out of a total 2260 HCWs working at our hospital were partially vaccinated with a single-dose BNT162b2 vaccine. Of these, 38 received the BNT162b2 vaccine elsewhere. An additional 25 HCWs were identified as receiving a single-dose ChAdOx1 vaccine and were excluded from the analyses. The split by age, sex, ethnicity and staff group is shown in Table 1. Uptake was higher for male (67.3%) than females (61.0%) ($p = 0.004$). Statistically significant differences were seen between ethnic groups with whites and Asians more likely to be partially vaccinated but only 28.5% of black and Afro-Caribbean and 45.0% of mixed-race staff being partially vaccinated ($p = 0.001$). Differences were also seen between staff groups with approximately half of nursing (55.5%) and clinical support (50.4%) staff, and even fewer portering, domestic and catering staff (31.6%) being partially vaccinated ($p = 0.001$).

The number of new cases identified at the RNOH since October 2020 is shown in Fig. 2. A total of 51 cases were identified, of which 2 were excluded due to a previous positive test within 90 days. Of the 49 new cases, 26 (53.1%) were unvaccinated; 20 symptomatic infection and 6 were asymptomatic, 16

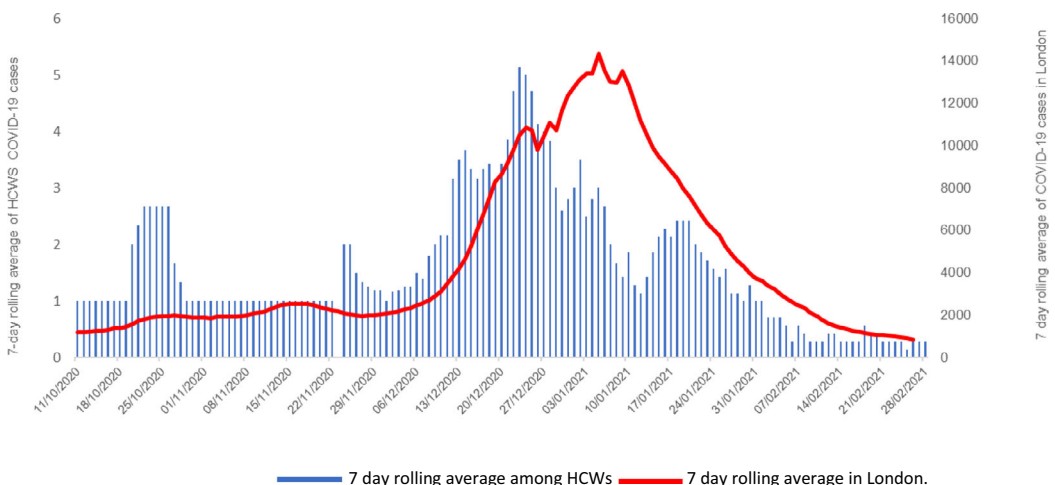

**Fig. 1 The 7-day rolling average of COVID-19 cases for the study hospital compared with the London regional average.** London date includes individuals with a least one positive COVID-19 test results (either laboratory reported or lateral flow device). The date of a new case defined by the specimen date. HCW cohort included cases with a positive laboratory COVID-19 test. HCWs with HCWs with a previous positive within 90 days were excluded. The date of a new case defined as date of symptom onset or specimen date if asymptomatic. The rolling 7-day average of confirmed cases in the London Region (red) was 14,326.6 on 5 January, decreasing to 843.4 on 26 February 2021. The rolling 7-day average of new cases in HCWs at the study hospital (blue) was 3.0 on 5 January, decreasing to 0.3 on 26 February 2021.

**Table 1 Comparing baseline characteristics in partially vaccinated HCWs receiving single-dose BNT162b2 vaccine and unvaccinated groups.**

| Baseline characteristic | Total, n | Partially vaccinated with BNT162B2, n (%) | Unvaccinated, n (%) | p value |
|---|---|---|---|---|
| Total | 2235 | 1409 (63.0) | 826 (37.0) | |
| *Sex, n (%)* | | | | |
| Male | 722 | 486 (67.3) | 236 (32.7) | 0.004 |
| Female | 1513 | 923 (61.0) | 590 (39.9) | |
| *Race or ethnic group* | | | | |
| White | 1109 | 805 (72.6) | 304 (27.4) | <0.001 |
| Asian | 595 | 405 (68.1) | 190 (28.5) | |
| Black or Afro-Caribbean | 382 | 109 (28.5) | 273 (71.5) | |
| Chinese | 19 | 13 (68.4) | 6 (31.6) | |
| Mixed | 60 | 27 (45.0) | 33 (55.0) | |
| Other | 70 | 50 (71.4) | 20 (28.6) | |
| *Age group, years* | | | | |
| 16–34 | 584 | 303 (51.9) | 281 (48.1) | <0.001 |
| 35–54 | 1234 | 811 (65.7) | 423 (34.3) | |
| >55 | 417 | 295 (70.7) | 122 (29.3) | |
| *Staff group* | | | | |
| Administrative and clerical | 629 | 461 (73.3) | 168 (26.7) | <0.001 |
| Nursing | 494 | 274 (55.5) | 220 (44.5) | |
| Allied health professionals | 240 | 188 (78.3) | 52 (21.7) | |
| Clinical support staff | 234 | 118 (50.4) | 116 (49.6) | |
| Surgeons and medics | 231 | 183 (79.2) | 48 (20.8) | |
| Portering and catering | 225 | 71 (31.6) | 154 (68.4) | |
| Professional scientific and technical staff | 182 | 114 (62.4) | 68 (37.4) | |

For gender, ethnicity, age group and staff group, Pearson's two-sided $\chi^2$ test was performed for hypothesis testing, without adjustment for multiple comparisons.

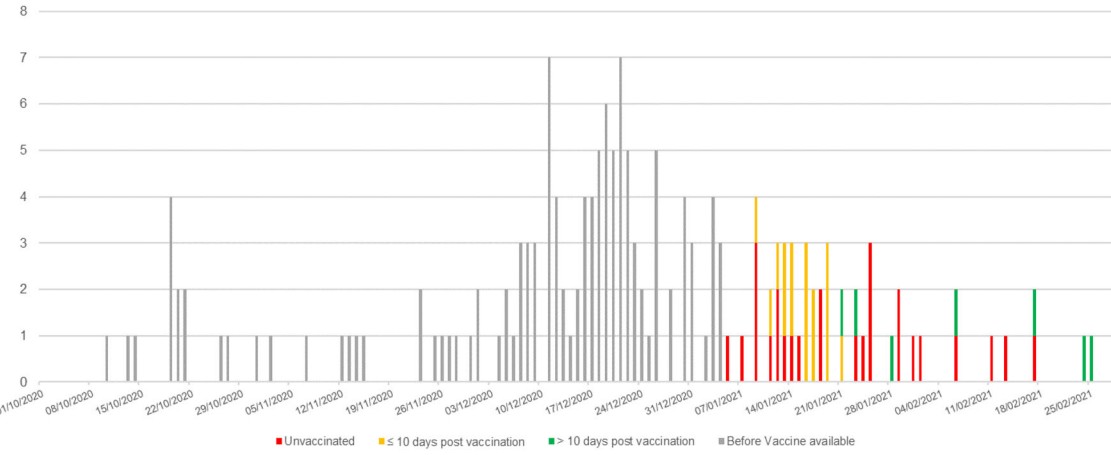

**Fig. 2 New cases of COVID-19 occurring more or less than 10 days after receiving a single-dose BNT162b2 vaccine.** A total of 49 new cases of COVID-19 were following the start of BNT162b2 vaccination offered at the onsite hospital vaccination centre over a 2-week period from 5 January 2021. Of the 49 new cases, 26 (53.1%) were unvaccinated (red); 16 (32.7%) were within 10 days of vaccination (orange); and 7 (14.3%) were beyond 10 days of vaccination (green).

(32.7%) were within 10 days of vaccination; 15 symptomatic infection and 1 was asymptomatic and 7 (14.3%) were beyond 10 days of vaccination; 6 were symptomatic and 1 was asymptomatic infection. Figure 3 shows the Kaplan–Meier curves of new cases identified in partially vaccinated and unvaccinated groups. The curves appear to be congruent from day 0, start to diverge at day 14, and then continue to diverge up to day 42. Separate Cox regression analyses performed for the two stages of the follow-up period, for partially vaccinated and unvaccinated groups, are compared in Table 2. There was no significant difference in risk of infection between partially vaccinated and unvaccinated groups for the first 13 days of follow-up. This was the case in both the unadjusted and adjusted analysis. For the second follow-up period from day 14

onwards, the adjusted analyses showed a 70.0% (95% CI 6.0–91.0%; $p = 0.04$) reduction in the risk of infection in the partially vaccinated group.

No serious adverse events were reported and we submitted no yellow cards to the MHRA.

From 16 February to 26 March, 214 (25.9%) of the 826 in the unvaccinated cohort were identified as receiving a COVID-19 vaccine. Of these, 154 HCWs received a first dose of BNT162b24 vaccination during the second period of vaccination at our hospital, 22 received a first dose of BNT162b24 elsewhere and 38 received a ChAdOx1 vaccine elsewhere. By 26 March, the total HCWs partially vaccinated with either BNT162b24 or ChAdOx1 vaccines increased from 63.5% (1434/2260) to 72.9% (1648/2260).

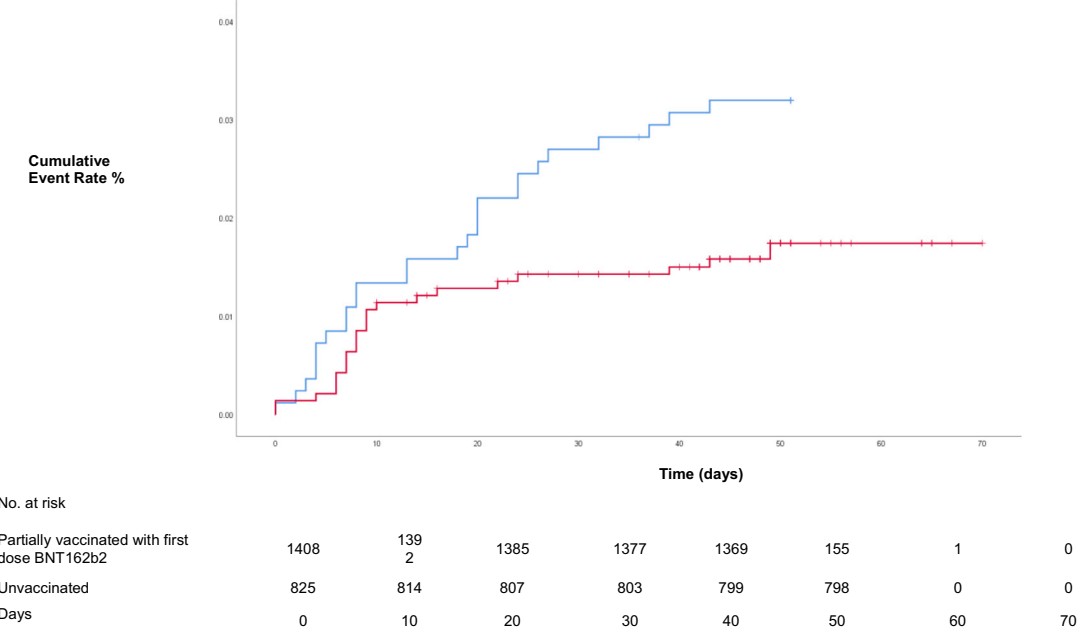

**Fig. 3 Kaplan–Meier estimates of time to first occurrence of COVID-19 in vaccinated and unvaccinated HCWs.** In all, 23 of 1408 from the partially vaccinated and 26 of 825 from the unvaccinated group were identified as new COVID-19 cases during the follow-up period. Up to day 13, there was no significant difference in the cumulative risk of new COVID-19 infection between both groups. From day 14, there was a significant reduction in the cumulative risk of new COVID-19 infection among partially vaccinated HCWs.

**Table 2 Comparison of time to COVID-19 infection in unvaccinated and partially vaccinated HCWs.**

| Follow-up period | Analysis[a] | COVID—unvaccinated n/N | COVID—partially vaccinated n/N | Hazard ratio (95% CI)[b] | p value |
|---|---|---|---|---|---|
| Up to day 13 | Unadjusted | 13/825 | 16/1408 | 0.72 (0.35, 1.50) | 0.38 |
| | Adjusted[c] | – | – | 0.8 (0.31, 2.47) | 0.80 |
| Day 14 onwards | Unadjusted | 13/812 | 7/1392 | 0.33 (0.13, 0.83) | 0.02 |
| | Adjusted[c] | – | – | 0.30 (0.09, 0.94) | 0.04 |

Cox hazard ratios and 95% confidence intervals are reported with corresponding p values. p values were estimated by the Wald test. Statistical significance was set at p < 0.05, and two-sided p values were used.
[a]1 HCW vaccinated and 1 unvaccinated HCW tested positive and within 90 days of vaccination were excluded from the analyses.
[b]Expressed as hazard of COVID-19 in partially vaccinated group relative to unvaccinated group.
[c]Analyses were adjusted for age, sex, staff group, ethnicity and underlying COVID-19 rate.

## Discussion

In our study partially vaccinated HCWs had a 70.0% reduction in risk of symptomatic and asymptomatic infection with a duration of up to 42 days. Overall uptake of a single dose of BNT162b2 vaccine was 62.3%; however, there were significant differences in uptake between groups.

Males were more likely to be partially vaccinated and the proportion of HCWs partially vaccinated increased with increasing age. A lower uptake was seen in nursing and clinical support staff, but the lowest rates were seen in portering, domestic and catering staff, all of whom are at risk of coming into close contact with patients with COVID-19 or their environment.

It was striking that uptake was lower amongst black and Afro-Caribbean staff and those of mixed heritage. This is concerning as these groups have been disproportionately adversely affected by COVID-19 and remain at risk[11]. In addition, these groups are over-represented amongst the staff groups above. Furthermore, London has a higher proportion of staff from these groups when compared with the rest of the UK that may have an impact on a health service level[12].

Whilst the uptake of first dose of BNT162b2 in our study is encouraging, it is likely that the current rate will not fully protect against nosocomial spread. On a population level it is unclear what proportion needs to be vaccinated to confer herd immunity. Several estimates have put the proportion between 70 and 80% across a population; however, hospital populations are dynamic with patients, often with relatives or carers, changing frequently, and staff often working across different sites[13]. Therefore, other infection prevention control measures, including screening, pre-admission isolation, social distancing and use of personal protective equipment, are likely to remain important in minimising risk of exposure in these settings.

It is clear that more work is needed to understand the reasons why uptake is lower in certain groups. It would be important to know whether this represents inequity of access or whether there is true hesitancy in receiving the vaccine and if so, what are the reasons why staff may not wish to be vaccinated. Some work has already begun around this nationally and we have conducted a survey and focus groups locally to explore potential reasons[14]. Commonly stated reasons included access issues; concern around immediate side effects, including allergy; and concern around long term effects, particularly around fertility and pregnancy. Disappointingly, other reasons included being anti-vaccination, belief in various conspiracy theories and COVID-19 denial and

lack of trust in pharmaceutical companies. Whilst some of these concerns may be able to be addressed, others may not and research on the most effective ways to increase uptake is required.

Once we had identified the groups with lower rates of partial vaccination we attempted to improve uptake using several measures. These included increasing availability of information, mass communications and direct contact through line managerial support, including non-English speakers. A further 25.9% of previously unvaccinated HCWs came forward suggesting that it is possible to significantly increase uptake; however, it is not clear which measures are the most impactful and it requires significant effort and resource. We managed to increase our overall uptake to 72.9% that may be important if herd immunity is an important factor in preventing transmission.

Our 7-day rolling average of cases mirrored the London 7-day rolling average. This is not surprising as our hospital is within the London area and suggests that a substantial proportion of cases amongst our staff reflect what is happening in the local community rather than nosocomial spread within our institution: local track and trace reports found that the majority of cases reported contact with a household case or with another HCWs at work during a break. It was thought that spread occurred when personal protective equipment was not worn in order to eat or drink, and this is consistent with what was happening at other hospitals in the region. We saw relatively fewer cases after Christmas and during the first 2 weeks of January and we postulate that this was for two main reasons: first, the onsite testing centre closed for a period over Christmas and New Year; second, we suspect there was an effect due to more staff than usual being on annual leave. Our rate appeared to rise, peak and fall slightly before the London rate, and we postulate that this may be due to local screening picking up a combination of pre-symptomatic cases earlier than they would otherwise have been detected in the community and asymptomatic cases that would otherwise have not been detected. Once vaccinations began a difference in the groups testing positive for COVID-19 emerged. Cases initially occurred relatively equally in both groups; however, the groups diverged by 14 days with almost all subsequent cases occurring in the unvaccinated group. This is similar to what was reported in the initial efficacy study and most likely reflects the development of an effective immune response in the partially vaccinated group[3]. PHE recommends enhanced surveillance for anyone developing COVID-19 10 days or more after receiving a vaccine. Based on the results of the original BNT162B2 vaccine study, this cutoff might be too early to detect true vaccine failure and consideration should be given to delaying the cutoff until 14 days or more have elapsed.

We found a 70.0% reduction in risk of both asymptomatic and symptomatic infection from 14 to 42 days in partially vaccinated compared with unvaccinated HCWs. Our findings are comparable to a single-centre HCWs cohort study performed in Israel reporting BNT162b2 vaccine effectiveness of 75% (95% CI 72–84) up to 28 days following a first dose of BNT162b2 vaccine[15]. A multicentre UK cohort study calculated BNT162b2 vaccine effectiveness to be at least 70% (95% CI 53–87%) 21 days after the first dose[16]. This large multicentre UK study included volunteer HCWs who may not be representative of UK HCWs more generally. Our cohort study provides unique data on the effectiveness of a first dose of BNT2b2 vaccine up to 42 days in a healthcare setting. This is reassuring as the UK National Vaccination Strategy has been to delay administration of the second dose of the vaccine to 90 days[17]. These findings suggest that this strategy may be considered more widely, particularly if global vaccine supplies fall short of demand.

Potential limitations of this study include its single-centre retrospective design. It is possible that some HCWs received a vaccine elsewhere and this was not reported to us. Significant efforts were made to obtain these data and we believe that the effect of missing data here would be minimal. We acknowledge that previous COVID-19 infection may reduce susceptibility to future infection. We did not test for the presence of pre-existing antibodies, which were not routinely available. We cannot exclude this influencing the study outcome, but we postulate that inclusion of potentially less susceptible individuals in the unvaccinated arm would be to make the vaccine appear more effective. It is reassuring that we demonstrated a significantly lower risk of COVID-19 infection in partially vaccinated HCWs. Larger studies are required to verify these findings. Our study was conducted in a period of high prevalence but reducing incidence by the end of the follow-up period. It is possible that this could have flattened the curves in the Kaplan–Meier analysis having the effect of making the vaccine appear less effective.

In conclusion, initial vaccination rates among HCWs were generally high although uptake was lower in certain groups. We have shown that it is possible to improve vaccine uptake rates and efforts should focus on this, despite it being resource intensive. The BNT162B2 vaccine is effective from 14 days post vaccination in a frontline clinical setting and protection continues beyond 21 days post first dose without a second dose being given.

## Methods

We assessed uptake of COVID-19 vaccination among HCWs at our tertiary orthopaedic hospital. This study was conducted during a surge when endemic prevalence rates per 100,000 population were 1023.7 on 5 January, decreasing to 61.4 on 26 February 2021[6]. At the time, we were treating COVID-19 patients in three wards and in an expanded intensive care unit. All HCWs were offered a first dose of the BNT162b2 vaccine at our onsite vaccination centre over a 2-week period from 5 January 2021[7]. There were no holiday periods during the time that we were vaccinating at our centre. The time period for vaccination was chosen to match the total number of HCWs with the constraints of storage requirements, freeze thaw times and the desire to avoid vaccine wastage. Our online booking system was accessible to all HCWs and was used to manage vaccination appointments. Literature advertising vaccinations was available in English and the three most common non-English languages used by HCWs. Online information on our vaccination programme, webinars and face-to-face conversations with line managers were used to improve awareness and inform HCWs before and during the vaccination period.

Vaccines were administered by trained vaccinators in accordance with the National Protocol or by patient specific direction[8]. At the time of vaccination the dose and batch number were recorded and uploaded to the National Immunisation and Vaccination System (NIVS). All HCWs were observed for a minimum of 15 min post vaccination in accordance with the MHRA temporary authorisation for the BNT162b2 vaccine and were asked to report any serious adverse events to the vaccination team or occupational health department. HCWs who received a COVID-19 vaccine elsewhere were asked to report the type of vaccine and date of administration to line managers.

Human resource records containing baseline characteristics including age, gender, ethnicity and job role were collected for all HCWs directly employed by our hospital. Contractor companies provided the same dataset for HCWs working onsite but not directly employed by our hospital. We identified all HCWs who received the vaccine using data submitted to NIVS. Occupational health staff managed suspected COVID-19 infections in accordance with National Track and Trace Guidelines[9].

Cases were identified using Occupational Health records. Cases were either symptomatic or asymptomatic or pre-symptomatic individuals identified through mandatory onsite screening and were included if they had a laboratory confirmed positive PCR test within 42 days of the end of the vaccination programme. Cases were excluded if they had a previous positive test within 90 days. All cases were routinely questioned regarding potential exposures, contacts and whether they had received a COVID-19 vaccine. Cases were categorised as either: occurring within 10 days of vaccination, or 10 days or longer after vaccination. A 10-day cutoff was chosen because it matched the criteria recommended by Public Health England (PHE) for enhanced COVID-19 surveillance[10]. We calculated a rolling 7-day average of new cases for our hospital for comparison with reported regional London data[6].

In March 2020, a vaccine hesitancy working group, including human resources, infection prevention and control, pharmacy and clinical psychology, implemented further interventions to improve vaccine uptake. These included: provision of additional information in the form of frequently asked questions and other resources in several languages; webinars hosted by BAME ambassadors; webinars addressing fertility and pregnancy concerns; support for booking for staff who had reduced access to the booking system; drop-in sessions facilitated by different staff groups; and one to one conversations with line managers.

Between 15 and 26 March, our hospital implemented a second period of BNT162b2 vaccinations. Unvaccinated staff not receiving a first dose during the initial 42-day period were screened to determine if they had received a COVID-19 vaccine either onsite or elsewhere.

The study was approved by The Royal National Orthopaedic Hospital Research and Innovation Committee (RNOH RIC). A waiver of consent for clinical data collection was granted due to the use of anonymised data.

**Reporting summary**. Further information on research design is available in the Nature Research Reporting Summary linked to this article.

## Data availability

The datasets generated during and/or analysed during the current study are available from the corresponding author on reasonable request.

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

## Acknowledgements

We would like to acknowledge the COVID-19 Vaccination Team at the Royal National Orthopaedic Hospital. We would also like to acknowledge Paul Bassett at Statsconsultancy Ltd for his support with the statistical analysis.

## Author contributions

T. A., S. W., J. A. S and T. W. R. B contributed to the study concept and design; T. A., M. H. and A. S. assisted with data collection; T. A. conducted the data analysis; T. A. and S. W. drafted the manuscript; T. A., S. W., J. A. S., M. H., A. S., I. H. and T. W. R. B. critically reviewed the final draft of the manuscript. All authors approved the final version to be published.

## Competing interests

The authors declare no competing interests.
