## [Peer Review File · Nature Communications]

Reviewers' Comments:

Reviewer #1:

Remarks to the Author:

Re: NCOMMS-21-06947

Experience of COVID-19 Vaccination of Healthcare Workers in a Hospital Setting

This is an interesting description of experience accumulated in a single medical center with regards to two questions: a) characteristics of vaccinated HCWs compared with unvaccinated . b) risk reduction in vaccinated compared with unvaccinated HCWs. Overall I find the methods satisfactory. The discussion needs to focus on study findings and interpretation (see comments below).

My main comments are:

L109: I presume all cases were reported from the onset of the pandemic. Did you include HCWs who recovered from earlier SARS-CoV-2 infection in your analyses? This may introduce bias because the risk of infection in recovered after receiving 1st dose Pfizer vaccine may be lower as this dose serves as a booster. Can you exclude this group?

L133: In other countries, by the end of 2020 risk of infection in hospital during patient contact was low, and community exposure (proportional to disease rates in the community) was a main determinant of infection risk in HCWs (and not direct patient contact). From eyeballing figure 1 it seems like the number of new infections in London was around 14000 on Jan 5th and 10000 on Jan 13th – this means exposure risk was changing. Can you conduct sensitivity analysis to check if this impacts your results? Other similar analyses adjusted to period of vaccination or community exposure rates as other similar analyses did (See Amit, Lancet 2021 and see Hall, Lancet 2021 preprint L222: https://papers.ssrn.com/sol3/papers.cfm?abstract_id=3790399)

Minor comments / suggestions:

Title: not very informative – suggest you describe your study (who where when what): for example: BNT162b2 Vaccine uptake and COVID-19 risk reductions in vaccinated HCWs – a single center cohort study, London, UK

Abstract L66: this is not the first report of Pfizer vax VE (see refs above and others)

L67, L94: Usually we define efficacy in clinical trials and effectiveness in real life use.

L98: ITU?

L98: Suggest use generic vaccine name.

L101: RNOH?

L101: Minimum dataset?

L107: How exactly is vaccination status documented in HCWs vaccinated outside hospital? Did some receive ChAdOx vaccine?

L156-7: Unclear sentence without results? Is this part of the discussion?

L149 and L158: 80.% For consistency use single decimal point (or whole percentage) across the manuscript and in confidence intervals.

Tables and figures: suggest you provide a title which reflects the data presented so tables and figures could stand alone: Who where when how many included. Our hospital is not informative.

Figure 3: Please provide number vaccinated and unvaccinated (per 10 days post vaccine) included and Cumulative number of events, to complete the KM curve.

Discussion:

Suggest in the first para provide your main new findings in context – risk reduction from single dose of Pfizer, with duration up to 50 days, and HCW groups with lower vaccine uptake. Then use 2-3 paragraphs to compare major findings with literature and finally conclude.

L183: herd immunity is not really something you examined and not a term used outside whole community setting. Can't see why mention this, certainly not in your first para. Nosocomial spread is prevented by vaccination and PPE.

L188: Show general wards have younger staff or provide reference. Why is this important.

L201-203: This study was not aimed to actively follow vaccine AEs and results of AE surveillance (if this was conducted) are not reported – why is this part of the discussion?

References – need to update and add real life data on Pfizer vaccine effectiveness from studies already published.

Reviewer #2:

Remarks to the Author:

This is a well written manuscript that attempts to address whether or not a single vaccine dose of the Pfizer/BioNTech vaccine would protect against symptomatic and asymptomatic infection in vaccinated healthcare workers at a single healthcare facility. Overall the findings demonstrate a clear efficacy of the vaccine beyond 14 days of immunization amongst all healthcare workers. The authors also demonstrate differences in the proportion of vaccinated individuals by sex, race/ethnicity, age, and occupation.

Major comments:

-The authors should more explicitly state that healthcare workers were not "fully vaccinated", meaning that they had only received one dose of the vaccine. Suggest using a term other than "vaccinated" for those who received one dose compared to those who received no doses (e.g. partially vaccinated versus unvaccinated).

-the authors should go into more depth about their vaccine delivery program in the methods and briefly discuss opportunities for improvement for equity with regards to vaccine access in their discussion. It appears as though the vaccine was only offered onsite for 8 days. Why only 8 days?

-suggest reviewing for typos and grammar (e.g. line 98: ITU instead of ICU).

Minor comments:

-Line 74: suggest stating "rates among HCW were generally high" instead of "good"

-Introduction: suggest mentioning that this study was conducted during the surge

-Methods: The authors briefly discuss vaccine eligibility and registration for receiving a vaccine, however more details should be given. Was there any prioritization given to different employees over others for registering to receive the vaccine? were they all invited at the same time? was there adequate supply of vaccine to immunize all of the healthcare workers? This should be mentioned in the methods as this might affect vaccine acceptance as some individuals may have more schedule flexibility to receive their vaccine compared to others.

-Methods: were there any vaccine education initiatives at the hospital to counter vaccine hesitancy and misinformation?

-Methods: how were individuals who had previously had COVID-19 handled in the mandatory surveillance testing program? Were they still required to undergo testing?

-Methods: was registration for vaccines conducted in English or in other languages as well? this might also impact vaccine acceptance? Additionally if registration was online only, did all staff have equal access to the internet to register for the vaccine?

-Results: it would be great if the authors could provide the number of asymptomatic and symptomatic cases of COVID-19. Were there any data on hospitalization in those who had COVID (realize the numbers are small but would still be very interesting)?

-Line 208: the authors should mention the number of days that the vaccine sites were closed during the holidays.

-Figure 2: suggest changing the colors for the graph as this may be difficult to read for those who can not differentiate between red and green.

Dear Mr Azamgarhi,

REVIEWER COMMENTS

Reviewer #1 (Remarks to the Author):

Re: NCOMMS-21-06947

Experience of COVID-19 Vaccination of Healthcare Workers in a Hospital Setting

This is an interesting description of experience accumulated in a single medical center with regards to two questions: a) characteristics of vaccinated HCWs compared with unvaccinated. b) risk reduction in vaccinated compared with unvaccinated HCWs. Overall I find the methods satisfactory. The discussion needs to focus on study findings and interpretation (see comments below).

My main comments are:

L109: I presume all cases were reported from the onset of the pandemic. Did you include HCWs who recovered from earlier SARS-CoV-2 infection in your analyses? This may introduce bias because the risk of infection in recovered after receiving 1st dose Pfizer vaccine may be lower as this dose serves as a booster. Can you exclude this group?

The reviewer raises a good point. We agree, HCWs with antibodies acquired through previous infection may be less susceptible to future infection. Unfortunately, we do not have accurate information on previous infection status so we cannot exclude these HCWs. We do not think this could be known for certainty unless all staff were regularly screened since the start of the start of the pandemic. We have acknowledged this point in the text in the discussion (L273).

L133: In other countries, by the end of 2020 risk of infection in hospital during patient contact was low, and community exposure (proportional to disease rates in the community) was a main determinant of infection risk in HCWs (and not direct patient contact). From eyeballing figure 1 it seems like the number of new infections in London was around 14000 on Jan 5th and 10000 on Jan 13th – this means exposure risk was changing. Can you conduct sensitivity analysis to check if this impacts your results? Other similar analyses adjusted to period of vaccination or community exposure rates as other similar analyses did (See Amit, Lancet 2021 and see Hall, Lancet 2021 preprint).

Thank you, we agree with the comment. This study was conducted during a period of high prevalence that was decreasing from the start of vaccination to the end of follow-up. This may reduce the chance of seeing a risk difference over time. This is an important point and we have subsequently re-run the analyses with an adjustment for community prevalence.

Kindly see L152 to 154 in the methods for amendments to the statistical analysis and Table 2 in the results.

L222: https://papers.ssrn.com/sol3/papers.cfm?abstract_id=3790399)

Thank you, we have included and referenced other real-world BNT162b2 vaccine studies as part of the discussion. Kindly see L262-270.

Minor comments / suggestions:

Title: not very informative – suggest you describe your study (who where when what): for example: BNT162b2 Vaccine uptake and COCOVID-19 risk reductions in vaccinated HCWs – a single center cohort study, London, UK.

We agree and have changed the title to “BNT162b2 Vaccine Uptake and Effectiveness in UK Healthcare Workers – A Single Centre Cohort Study”

Abstract L66: this is not the first report of Pfizer vax VE (see refs above and others)

Thank you. We have altered the wording due to “Several COVID-19 vaccines against SAR-CoV-2 have demonstrated high efficacy in clinical trials. This early report describes their use in a healthcare setting.” Kindly see L62/63

L67, L94: Usually we define efficacy in clinical trials and effectiveness in real life use.

Thank you. Changed in both instances.

L98: ITU?

A UK term, apologies. This has been changed to ICU. Kindly see L98-99.

L98: Suggest use generic vaccine name.

Changed to BNT162b2 throughout the manuscript. Thank you.

L101: RNOH?

The Royal National Orthopaedic Hospital (RNOH). Apologies this abbreviation was not defined. This has been removed as it is not necessary. We refer to our hospital

L101: Minimum dataset?

This term has been defined specifically. Kindly see L108.

L107: How exactly is vaccination status documented in HCWs vaccinated outside hospital? Did some receive ChAdOx vaccine>

This is an excellent point. We were initially not aware of reports of staff receiving the vaccine elsewhere. To address this, we reviewed data obtained within our hospital at the end of March. This showed that 38 HCWs received the BNT162b2 elsewhere before, during and up to 42 days after the start of vaccinations. We decided to re-run the statistical analysis to include all 38 HCWs. A further 25 HCWs received the ChAdOx vaccine elsewhere either before, during or up to 42 after our vaccination

programme. Due to the small numbers, we excluded these HCWs from the analysis. We can confirm that none of the additional staff that were vaccinated elsewhere developed COVID infection after 10 days following vaccination.

The methods (L111-112), results (L161-163).

We believe that there is no missing data however there is still the possibility that some HCWs may not have reported that they received a COVID-19 vaccine elsewhere. We have added a line in the limitations section of the discussion to state this. Kindly see (L272– 273).

L156-7: Unclear sentence without results? Is this part of the discussion?

Thank you, this sentence has been removed.

L149 and L158: 80.% For consistency use single decimal point (or whole percentage) across the manuscript and in confidence intervals.

Thank you, the manuscript has been changed.

Tables and figures: suggest you provide a title which reflects the data presented so tables and figures could stand alone: Who, where when how many included. Our hospital is not informative.

Thank you. Titles have been updated.

Figure 3: Please provide number vaccinated and unvaccinated (per 10 days post vaccine) included and Cumulative number of events, to complete the KM curve.

Please see the revised Figure 3 complete with table of numbers at risk by group and cumulative numbers of events per 10 time period. After adding the data it became apparent that the number of HCWs who were still at risk in the partially vaccinated group reduced significantly. Therefore we extended the follow up period so that the 42 days started on the last day of vaccinations. We then repeated the full analysis.

Discussion:

Suggest in the first para provide your main new findings in context – risk reduction from single dose of Pfizer, with duration up to 50 days, and HCW groups with lower vaccine uptake. Then use 2-3 paragraphs to compare major findings with literature and finally conclude.

Thank you. We have changed the start of the discussion based on your recommendation. Kindly see L214-216.

L183: herd immunity is not really something you examined and not a term used outside whole

community setting. Can't see why mention this, certainly not in your first para. Nosocomial spread is prevented by vaccination and PPE. –

Thank you. This section has been moved and reworded base on your comment. Kindly see L224-229.

L188: Show general wards have younger staff or provide reference. Why is this important?

We do not have data to back up that statement and have removed it based on your comment.

L201-203: This study was not aimed to actively follow vaccine AEs and results of AE surveillance (if this was conducted) are not reported – why is this part of the discussion?

References – need to update and add real life data on Pfizer vaccine effectiveness from studies already published.

Good point, and on reflection this is not the focus of the study. We have replaced this section with discussion of our findings and how they compare with other real-life data from studies. Kindly see L262-270. All referenced.

Reviewer #2 (Remarks to the Author):

This is a well written manuscript that attempts to address whether or not a single vaccine dose of the Pfizer/BioNTech vaccine would protect against symptomatic and asymptomatic infection in vaccinated healthcare workers at a single healthcare facility. Overall the findings demonstrate a clear efficacy of the vaccine beyond 14 days of immunization amongst all healthcare workers. The authors also demonstrate differences in the proportion of vaccinated individuals by sex, race/ethnicity, age, and occupation.

Major comments:

-The authors should more explicitly state that healthcare workers were not "fully vaccinated", meaning that they had only received one dose of the vaccine. Suggest using a term other than "vaccinated" for those who received one dose compared to those who received no doses (e.g. partially vaccinated versus unvaccinated).

This is an important point and we have changed every instance where vaccinated has been stated too partially vaccinated. We have also clarified the dosing schedule in the introduction. Kindly see lines 91-92. "The dosing schedule for the Pfizer vaccine is a single 30mcg dose at day 0 and repeated at a minimum of 21 days but due to limited availability, UK government strategy has been to delay 2nd dose until 3 months".

-the authors should go into more depth about their vaccine delivery program in the methods and

briefly discuss opportunities for improvement for equity with regards to vaccine access in their discussion. It appears as though the vaccine was only offered onsite for 8 days. Why only 8 days?

The time period for vaccination was chosen to match to the total number of HCWs with the constraints of storage requirements, freeze thaw times and the desire to avoid vaccine wastage. Kindly see L100-101.

-suggest reviewing for typos and grammar (e.g., line 98: ITU instead of ICU).

Thank you. The ICU is referred to as the ITU in the UK! Intensive Treatment Unit. We have adopted the global term. The manuscript has been reviewed in detail for typos and grammar.

Minor comments:

-Line 74: suggest stating "rates among HCW were generally high" instead of "good"

Thank you. This has changed. Kindly see L72. Also changed L281 in the conclusions.

-Introduction: suggest mentioning that this study was conducted during the surge when endemic prevalence rates of cases per 100,000 were...

Yes. Good point. The study was conducted at time of high community prevalence which we now state. We have stated the prevalence rates.

Kindly see L93-94 in the introduction and L97-98 in the methods.

-Methods: The authors briefly discuss vaccine eligibility and registration for receiving a vaccine, however more details should be given.

- Was there any prioritization given to different employees over others for registering to receive the vaccine?

Thank you for this comment. No prioritisation was applied to register for vaccination as there was enough vaccine for all HCWs over the 2-week vaccination period. Kindly see L100-102. "The time period for vaccination was chosen to match to the total number of HCWs with the constraints of storage requirements, freeze thaw times and the desire to avoid vaccine wastage."

- Were they all invited at the same time?

Yes, all staff were sent an SMS on Christmas Eve which is the time we were certain that we would get delivery of vaccine.

- Was there adequate supply of vaccine to immunize all the healthcare workers?

Yes, we matched the number of HCWs working at our institution to the maximum number of doses that could be drawn up from of once outer carton (1170) to ensure there was potential to vaccinate all HCWs.

This should be mentioned in the methods as this might affect vaccine acceptance as some individuals may have more schedule flexibility to receive their vaccine compared to others.

We had adequate vaccine supplies for all HCWs. The time of the slots were spread across the day so that vaccine slots were available for staff working day and night shift patterns.

-Methods: were there any vaccine education initiatives at the hospital to counter vaccine hesitancy and misinformation?

Yes and we have provided more information on the vaccine education initiatives implemented before we offered 1st dose vaccines in January. Kindly see L103-105.

We have also included information on the further vaccine hesitancy work conducted in March 2021 and calculated the number of previously unvaccinated HCW's who came forward after specific measures to improve vaccine uptake were implemented.

Kindly see line L125-133 in the methods, L181-185 of the results and L238-243 of the discussion.

-Methods:

- How were individuals who had previously had COVID-19 handled in the mandatory surveillance testing program?

Thank you for the comment. All staff with or without previous infection were required to continue with fortnightly surveillance testing.

- Were they still required to undergo testing?

Yes, as described above.

-Methods: was registration for vaccines conducted in English or in other languages as well? this might also impact vaccine acceptance? Additionally, if registration was online only, did all staff have equal access to the internet to register for the vaccine?

Both important points, thank you for raising these.

Kindly see L103-106. "Our online booking system was accessible to all HCWs and was used to manage vaccination appointments. Literature advertising vaccinations was available in English and the 3 most common non-English languages used by HCWs. Online information on our vaccination programme, webinars, and face-to-face conversations with line managers were used to improve awareness and inform HCWs before and during the vaccination period. "

-Results: it would be great if the authors could provide the number of asymptomatic and symptomatic cases of COVID-19. Were there any data on hospitalization in those who had COVID (realize the numbers are small but would still be interesting)?

We agree and it is an excellent point. 42 of 49 were symptomatic at the time of identification. We cannot exclude the possibility that they were pre-symptomatic. We do not have data on hospitalisations.

-Line 208: the authors should mention the number of days that the vaccine sites were closed during the holidays.

Thank you. We have clarified this in the manuscript. Kindly see L100.

-Figure 2: suggest changing the colours for the graph as this may be difficult to read for those who cannot differentiate between red and green.

We would be happy to change the colours to whatever the editorial team recommend.

Reviewers' Comments:

Reviewer #1:

Remarks to the Author:

Manuscript was revised according to my suggestions

Reviewer #2:

Remarks to the Author:

The authors have done an excellent job responding to the reviewer's suggestions.

Minor suggestions

-line 153- should be Hazard ratios I believe

-line 171-172- suggest stratifying the symptomatic and asymptomatic by vaccine status

-line 272- would also mention that previously positive hcw could have tested positive on subsequent testing since individuals can test positive on repeat testing despite no longer being infectious, especially if your hospital policy was not to exclude them from subsequent testing.

Dear Mr Azamgarhi,

REVIEWERS' COMMENTS

Reviewer #1 (Remarks to the Author):

Manuscript was revised according to my suggestions

Reviewer #2 (Remarks to the Author):

The authors have done an excellent job responding to the reviewer's suggestions.

Minor suggestions

-line 153- should be Hazard ratios I believe.

Good spot, thank you. Changed.

-line 171-172- suggest stratifying the symptomatic and asymptomatic by vaccine status

Thank you, that is a good suggestion and we have made changes to that section to stratify symptomatic and asymptomatic by vaccine status. Kindly see L174 to 177.

-line 272- would also mention that previously positive hcw could have tested positive on subsequent testing since individuals can test positive on repeat testing despite no longer being infectious, especially if your hospital policy was not to exclude them from subsequent testing.

Thank you, we excluded positive cases that had a positive test within 90 days of being vaccinated. There were 51 cases were identified during follow-up, and 2 were excluded because they did not meet the definition of a new case. Of the two excluded, 1 HCW was vaccinated and 1 HCW was unvaccinated. Although they were excluded from the number of cases (the numerator), they were not excluded from the number at risk in each group (the denominator). We have re-run the analysis and there were no changes to the results. We have updated the Figure 3 with the Kaplan Meier analysis.

We are very thankful to the reviewer for pointing this out. We have made changes to clarify the exclusion of cases with a positive test within 90 days. Kindly see in the methods L121-122, included all cases and exclusions in the results L170-172 and updated the discussion as per the reviewer's comment. Please see L273 – 274.